# Postpartum Depression in COVID-19 Days: Longitudinal Study of Risk and Protective Factors

**DOI:** 10.3390/jcm11123488

**Published:** 2022-06-17

**Authors:** Hadar Gluska, Noga Shiffman, Yael Mayer, Shiri Margalit, Rawan Daher, Lior Elyasyan, Maya Sharon Weiner, Hadas Miremberg, Michal Kovo, Tal Biron-Shental, Liat Helpman, Rinat Gabbay-Benziv

**Affiliations:** 1Obstetrics and Gynecology, Meir Medical Center, Kfar Saba 4428164, Israel; hadargluska@gmail.com (H.G.); swmaya@gmail.com (M.S.W.); michalkovo@gmail.com (M.K.); tal.biron-shental@clalit.org.il (T.B.-S.); 2Sackler Faculty of Medicine, Tel-Aviv University, Tel-Aviv 6997801, Israel; shiri5mm@gmail.com (S.M.); dasile2@gmail.com (H.M.); 3Department of Psychiatry, Rambam Medical Center, Haifa 3525408, Israel; nogashiffman@gmail.com; 4The Ruth and Bruce Rappaport Faculty of Medicine, Technion-Israel Institute of Technology, Haifa 3200003, Israel; rawan.med91@gmail.com (R.D.); liorhanoh@gmail.com (L.E.); 5Department of Counseling and Human Development, Faculty of Education, University of Haifa, Haifa 3498838, Israel; yaelmayer10@gmail.com (Y.M.); liat.helpman@gmail.com (L.H.); 6Obstetrics and Gynecology, Hillel Yaffe Medical Center, Hadera 7404703, Israel; 7Obstetrics and Gynecology, Edith Wolfson Medical Center, Holon 5822012, Israel; 8The Psychiatric Research Unit, Tel Aviv Sourasky Medical Center, Tel Aviv-Yafo 6423906, Israel

**Keywords:** post-partum depression, COVID-19, birth, EPDS

## Abstract

COVID-19 impacted the childbirth experience and increased the rates of postpartum depression (PPD). We assessed the longitudinal effects of the pandemic on the rates of PPD and evaluated the PPD causes and symptoms among women who delivered during the first COVID-19 quarantine in Israel. The participants completed online questionnaires 3 (T1) and 6 months (T2) following delivery. We used the ‘COVID-19 exposure’ questionnaire, while PPD symptoms, situational anxiety, and social support were evaluated with the EPDS, STAI, and MSPSS questionnaires. The mean EPDS scores increased between T1 and T2 (6.31 ± 5.6 vs. 6.92 ± 5.9, mean difference −0.64 ± 4.59 (95% CI (−1.21)–(−0.06)); t (244) = −2.17, *p* = 0.031), and the STAI scores decreased (45.35 ± 16.4 vs. 41.47 ± 14.0, t(234) = 4.39, *p* = 0.000). Despite the exposure to an increased number of COVID-19 events (3.63 ± 1.8 vs. (6.34 ± 2.3)), the impact of exposure decreased between T1 and T2 (8.91 ± 4.6 vs. 7.47 ± 4.1), *p* < 0.001). In the MSPSS, significant differences were noted on the family scale between the T1 (6.10 ± 1.3) and T2 (5.91 ± 1.4) scores; t (216) = 2.68, *p* = 0.0008. A regression analysis showed three statistically significant variables that correlated with increased EPDS scores: the MSPSS family subscale (F (1212.00) = 4.308, *p* = 0.039), the STAI scores (F (1212.00) = 31.988, *p* = 0.000), and the impact of exposure to COVID-19 (F (1212.00) = 5.038, *p* = 0.026). The rates of PPD increased for women who delivered during the first COVID-19 lockdown. Further research is warranted to help reduce PPD among these women.

## 1. Introduction

The global population dealt with unprecedented social, economic, and healthcare challenges during the COVID-19 pandemic. In terms of mental health, the COVID-19 pandemic markedly impacted the population worldwide, with an increase in the prevalence of anxiety, stress, and depression [1,2,3].

Compared to men, women are more susceptible to mental health adversities [4,5,6], and pregnant and post-partum women are especially vulnerable [7,8]. After birth, both familial and hormonal changes have a substantial impact on women’s mental health, which may aggravate existing psychopathologies or cause the development of specific psychopathology such as postpartum depression (PPD) [9].

More than a year after the initial outbreak, it is now clear that the pandemic has considerably affected all aspects of the parturients’ lives [10]. Social distancing, repeated government-enforced lockdowns, financial repercussions, job loss, furloughs, high levels of fear of COVID-19 infection, and the use of personal protective equipment during labor have all deeply affected the childbirth experience and have negatively impacted the mental wellbeing of postpartum women [11,12,13,14,15,16].

Pregnant women experienced higher levels of fear of COVID-19 infection [17], which were further augmented by the fear of endangering the fetus [18]. Additionally, social distancing and lockdowns have led to limited maternal social support networks, as well as limited access to health care services. This, in turn, has increased the maternal risk for the development of psychological disorders [15,19]. Moreover, the use of personal protective equipment was found to be an independent factor for developing depressive and post-traumatic stress symptoms during the postpartum period [20].

Thus, it is not surprising that most studies demonstrated an increase in the rates of postpartum depression (PPD) and other birth-related psychopathologies worldwide [21,22,23,24,25,26].

Given its global spread, society’s response to the COVID-19 pandemic has dynamically evolved, with variable effects on all areas of life. Gradually, and with accumulating data, the debilitating fear of the unfamiliar virus that paralyzed society decreased and made room for a sense of “desensitization” and adaptation to life under the pandemic [27]. This change has been observed at all levels—from the individual to medical authorities and even government guidelines. On the other hand, since many countries have been under restrictions and regulations for almost a year, the economic, social, and psychological difficulties deepened [28,29].

To date, numerous studies have addressed the effects of COVID-19 on PPD; however, most evaluated the level of symptoms during the pandemic at a specific point in time [21,22,23,24,25,26]. The social, medical, and evolving gubernatorial implications of the worldwide pandemic warrant an assessment of the longitudinal impact of the pandemic on the childbirth experience and PPD. Therefore, in this study, we aimed to evaluate the causes and symptoms of PPD across the six-month postpartum period among women who delivered during the first COVID-19 lockdown in Israel.

## 2. Materials and Methods

This was a longitudinal, multicenter, cohort analysis that was conducted in three medical centers in Israel and included women who gave birth between 10 March 2020 and 9 May 2020, during the first COVID-19 pandemic lockdown. The first evaluation (T1) was conducted approximately 10 weeks after childbirth, and the second (T2) was conducted 6 months after childbirth, during the second lockdown period, between 12 October and 8 December 2020. The first strict lockdown in Israel was from March to May 2020, while knowledge regarding the disease was still scant. With a decrease in the number of newly infected patients, the lockdown was gradually relaxed, and people resumed their daily routines with a common belief that “COVID-19 was over”. Schools reopened on 3 May, and almost all restrictions were canceled. Over time, the spread of COVID-19 increased again, which was followed by the reinstatement of restrictive regulations up to a second complete lockdown that began on 18 September.

Three centers were included in our study: Hillel Yaffe Medical Center (HYMC, Hadera, Israel), Meir Medical Center (MMC, Kefar Sava, Israel), and Wolfson Medical Center (WMC, Holon, Israel), all of which are university-affiliated facilities. The study was approved by each medical center’s Institutional Review Board (HYMC-20-0079, MMC-0169-20, WMC-143-20, NIH NCT04609501). Our primary aim was to assess the trends in PPD symptoms throughout the COVID-19 pandemic era over a 6-month period following childbirth, utilizing the Edinburgh Postnatal Depression Scale (EPDS), and to evaluate the associated risk and protective factors.

### 2.1. Study Participants

Women were eligible to participate following a live-birth delivery in one of the three participating medical centers during the aforementioned period, which was during the first lockdown enforced by the Israeli government. Parturients younger than 18 years or who delivered earlier than 34 gestational weeks were excluded. During T1, a comprehensive team of physicians and medical students, both Hebrew- and Arabic-speaking, approached all women by phone approximately 10 weeks after delivery—the accepted time interval from birth that is used to assess PPD or other birth-related psychopathologies [30]. Women were given a brief explanation of the study protocol and were invited to participate in the study. Following oral consent, a text message was sent to each of them, containing a link to an online questionnaire, either in Hebrew or in Arabic, according to the participant’s preference. The questionnaires were sent using the online Qualtrics survey platform. Each participant completed a written consent form at the beginning of the questionnaire in accordance with the specifications of the Helsinki Committee. Only participants who completed at least 70% of the T1 questionnaire were included in T2, which was conducted about 6 months later. Women were contacted once again, invited to participate in T2 (following the same protocol as in T1), and were asked to complete a similar online questionnaire.

### 2.2. Data Collection

Maternal demographics, obstetric history, pregnancy surveillance, labor and delivery data, and short-term maternal and neonatal outcome (until home discharge) were all retrieved from the comprehensive computerized perinatal database at each medical center. At all centers, the data were routinely collected at the time of admission to the labor ward, during labor and delivery, and at postpartum admission, and they were retrospectively retrieved and analyzed. Maternal data included previous medical history, medications including any psychiatric medications, smoking, etc. Obstetric characteristics included parity, previous obstetric history, and current pregnancy follow-up (1st- and 2nd-trimester genetic screening, anatomy scan, glucose status, any hypertensive disorders). Parameters regarding the course of labor and birth outcome were also included. In addition, variables concerning the postpartum course of both the mother and the newborn were collected, such as the number of hospitalization days, admission to a maternal or neonatal intensive care unit, and more.

### 2.3. Online Questionnaires

Using the Qualtrics survey platform, women were asked to provide demographic, socioeconomic, and obstetrical information and to complete a battery of mental health questionnaires.

*The demographic and socioeconomic details* included questions regarding ethnicity, religious tendencies, education, family status, employment status, and average household income.

*Fear of COVID-19* was measured using the validated Fear of COVID-19 Scale [31], which was found to be associated with anxiety, stress, and depression in the general population. The Hebrew version of the questionnaire was also validated [32]. The questionnaire includes seven statements such as “I am afraid of losing my life because of the Coronavirus”. Participants were asked to rate their degree of agreement with the statements on a five-point Likert scale of 1–5 (total score 7–35; higher score indicates more fear). The questionnaire has a good internal validity (Cronbach’s alpha 0.82), and in the current study, the scale showed an internal consistency of 0.84.

*The COVID-19 exposure questionnaire* was compiled to detect COVID-19-related life events. It includes 14 items (for example, “I was in contact with someone who was infected by the Coronavirus”). Each participant was asked to indicate whether she experienced such an event and to what extent it impacted her (on a three-point scale of 0–2). The number of events was summed for the “Exposure to COVID-19 events” variable (score 0–14; higher score indicates more exposure) and for the total impact of the events—the “Impact of Exposure to COVID-19 events” (score 0–28; higher score indicates that the events had a greater impact). We further asked the participants if they were concerned about their financial state following the COVID-19 pandemic, on a scale from 1 to 5 (1—“not worried at all”, 5—“worried all the time”). We called this variable “Financial difficulties due to the COVID19 pandemic”. This variable aimed to reflect the parturient’s perception of her economic issues (higher score indicates more financial difficulties), meaning the “economic preoccupation”.

*PPD symptoms* were evaluated using the validated EPDS questionnaire [33]. This questionnaire is composed of 10 items, scored using a four-point Likert scale (0–3; higher score reflects the severity of PPD symptoms), while a score of ≥10 is considered a possible diagnosis of depression. The reliability of this scale was found to be 0.88, and the standardized alpha coefficient was found to be 0.87 [34].

*Current anxiety* was evaluated using the validated six-item State Anxiety Inventory (STAI) questionnaire. This is a self-report questionnaire which, in its shortened form, included 6 of the 20 original items, presenting feelings that characterize anxiety. The participants are asked to rate the extent to which each of the feelings described characterized them in the last month on a four-point Likert scale (1–4). The total score is obtained by the summation of the items (final total scale 20–80); a higher score indicates higher levels of anxiety [35]. The reliability coefficient for this questionnaire was alpha = 0.82.

*Social support* experienced by women during this period was measured using the Multidimensional Scale of Perceived Social Support (MSPSS) [36]. This is a 12-item self-report scale that consists of three subscales including support by one’s significant other, family, and friends. Each subscale is represented by four items—for example: “There is a special person who is around when I am in need”. The participants were requested to respond using a seven-point Likert scale ranging from 0 to 6 (0—“very strongly disagree”, 6—“very strongly agree”). A higher score indicates greater subjective social support (final total range 1–7). The questionnaire has a good internal validity (Cronbach’s alpha 0.88), and in the current study, the scale showed an internal consistency of 0.92.

Ultimately, all the collected demographic and obstetric data were verified with each center’s perinatal and postnatal database.

For the Arabic version, previously translated and validated questionnaires were used, or, alternatively, questionnaires were translated and back-translated by native Arabic speakers. A complete set of the questionnaires is available as a Appendix A.

### 2.4. Data and Statistical Analysis

The final analysis included only women who answered at least 70% of the questionnaires in T1 and T2. The data analysis was performed with the SPSS v23.0 package (IBM Corp., Chicago, IL, USA). A paired T-test was conducted to compare the results of EPDS as well as other T1 and T2 questionnaire scores. Significant contributors that potentially affected the EPDS scores were assessed utilizing a repeated measures (within-subjects) ANOVA with a Greenhouse–Geisser correction.

Differences were considered significant when the *p*-value was less than 0.05.

## 3. Results

### 3.1. Study Population

Overall, 1462 women were delivered during the study period at the three participating medical centers. During T1, 1079 (74%) were contacted by phone. In total, 774 (53%) consented to answering the online questionnaires, and 429 (29%) answered over 70% of the questionnaires [19] and were later re-approached at T2. Among them, 377 (87.9%) gave their consent, and 246 completed at least 70% of the T2 questionnaires and thus entered the analysis (Figure 1). No between-group differences were found in the mean EPDS scores. (T2 dropouts: 5.70 ± 5.1 vs. T2 participants: 6.31 ± 5.6, *p* = 0.342). The participants that completed the T2 questionnaire were significantly older (T2 dropouts: 30.73 ± 5.5, T2 participants: 32.14 ± 5.1, *p* = 0.003) and were less economically preoccupied (T2 dropouts: 3.08 ± 1.2, T2 participants: 2.8 ± 1.2, *p* = 0.013) than their counterparts. The mean delivery to the T1 and T2 questionnaire response intervals was 11.03 ± 1.6 weeks and 28.48 ± 2.1 weeks, respectively.

For the entire cohort, the mean maternal age at delivery was 32.14 ± 5.1 years. The sample was representative of the Israeli population in its ethnic diversity (31): Three-quarters of the cohort (186/246, 75.6%) were Jewish and one-quarter was Arab (60/246, 24.4%), which is representative of the population ratio in Israel [37]. Overall, 229 (93.1%) of the participants were married or in a relationship, and for 65 (26.4%), this was their first delivery. The mean gestational age at delivery was 39.5 ± 1.1 weeks, and the mean neonatal birthweight was 3286.86 ± 11.1 g. Most women (198/246, 80.5%) delivered vaginally, and the rest (48, 19.5%) delivered by a cesarean section. The demographic, socioeconomic, and obstetric data, as well as the questionnaire mean scores of the study cohort, are presented in Table 1.

### 3.2. PPD Symptoms and COVID-19 Related Questionnaires

A paired sample t-test was conducted to compare the questionnaire scores between T1 and T2 (Table 2). Significant differences were found between the EPDS scores of T1 (6.31 ± 5.6) and T2 (6.92 ± 5.9) (mean difference −0.64 ± 4.59 (95%CI −1.21–0.06); t (244) = −2.17, *p* = 0.031). Fifty-seven women (23.2%) had high EPDS scores (≥10 points) in T1, compared with sixty-six women (26.8%) in T2.

As for COVID-19 related questionnaires, significant differences were found in the exposure to COVID-19 events in T1 (3.63 ± 1.8) and T2 (6.34 ± 2.3) (t(245) = −16.98, *p* = 0.000). Despite the increased exposure to COVID-19 events, the impact of exposure decreased (8.91 ± 4.6 from T1 vs. 7.47 ± 4.1 in T2; mean difference 1.45 ± 4.91; t(245) = 4.62, *p* = 0.000). No differences were found between the fear of COVID-19 in T1 (17.25 ± 5.3) and that in T2 (17.10 ± 5.3; t(244) = 0.42, *p* = 0.676), nor between the financial difficulties due to the COVID-19 pandemic in T1 (2.80 ± 1.2) and that in T2 (2.83 ± 1.1; t(266) = −0.581, *p* = 0.561). Figure 2 demonstrates these differences.

### 3.3. PPD Symptoms and Other Control Variables

Unlike PPD symptoms, the levels of situational anxiety, evaluated by STAI scores, decreased over time from 45.35 ± 16.4 at T1 to 41.47 ± 14.0 at T2 (t(234) = 4.39, *p* = 0.000). As for the MSPSS questionnaire, no differences were found in the total score between T1 (5.99 ± 1.1) and T2 (5.88 ± 1.2; t(219) = 1.87, *p* = 0.062); however, in the family-specific subscale, a significant decrease was noted between T1 (6.10 ± 1.3) and T2 (5.91 ± 1.4; scores: t(216) = 2.68, *p* = 0.008). Other MSPSS subscales’ scores did not differ significantly between T1 and T2.

Lastly, to better comprehend the differences between the EPDS scores in T1 and T2, we conducted a repeated measures (within-subjects) ANOVA with a Greenhouse–Geisser correction. The variables included in this analysis were those that were found to have significant differences between T1 and T2 (STAI scores, MSPSS family sub-scale, exposure to COVID-19 events, and the impact of exposure to COVID-19 events). Of these, the three variables that remained statistically significant and impacted the differences between the EPDS scores at T1 and T2 were the MSPSS family subscale (F(1212.00) = 4.308, *p* = 0.039), the STAI scores (F(1212.00) = 31.988, *p* = 0.000), and the impact of exposure to COVID-19 events (F(1212.00) = 5.038, *p* = 0.026) (Table 3).

## 4. Discussion

In this study, we aimed to evaluate PPD symptoms and their trajectory over 6 months among women who delivered during the first COVID-19 lockdown period. Unlike a natural decrease in PPD rates during normal times [38], our principal findings demonstrated that women experienced increased rates of PPD symptoms at 6 months postpartum compared to those at 3 months postpartum. This remained true when evaluating the EPDS total score (6.31 ± 5.1 vs. 6.95 ± 5.9) and also when comparing the number of women with high EPDS (≥10 points): 66 parturients (26.8%) in T2 compared to 57 (23.2%) in T1.

Our results are in concordance with previous studies, suggesting an absolute increase in the prevalence of PPD symptoms among parturients who gave birth during the COVID-19 pandemic (15–20). López-Morales et al. followed pregnant women during 50 days of quarantine and found that all women showed a gradual increase in psychopathological indicators and a decrease in positive affect [39]. Nevertheless, none of these previous studies demonstrated the longitudinal influence of the COVID-19 pandemic on the parturient’s mental health over a 6-month postpartum period.

PPD is a multi-factorial condition. Aside from the hormonal and socially significant changes that occur with every delivery, COVID-19 introduced a new framework which included all known factors as well as new factors. On one hand, with the ongoing pandemic, stress, financial worries, and general fear and anxiety may increase. On the other hand, because the crisis was global, increased knowledge and the gradual adaptation of all systems, including governments, medical health authorities, and even family and friends, allowed life to get back on track [40].

In our study, when evaluating confounders, potential contributors, and PPD symptoms, we found that the exposure to COVID-19 events was higher in T2, which is an expected finding, as COVID-19-related life events continue to increase as time passes by. Interestingly, the psychological impact of the exposure to COVID-19 decreased in T2 compared with that in T1, suggesting a desensitization to each COVID-19 event with time [27]. STAI was found to be lower in T2 than that in T1, and as for the MSPSS questionnaire, significant differences between T2 and T1 were found in the family subscale: the family support perceived by the parturients was significantly lower in T2 compared with the perceived family support in T1.

These three components (impact of COVID-19, state anxiety, and family support) are potential determinants that may explain the change in EPDS scores over time.

It seems that as the COVID-19 pandemic progresses over time, the effect on the maternal mental health of women who gave birth during this period did not disappear and even worsened.

Despite a lower impact of the exposure to COVID-19 and lower situational anxiety (measured by the STAI questionnaire), the prevalence and severity of PPD symptoms in T2 were greater compared to those in T1. This difference could not be explained by financial difficulties or the fear of COVID-19, which were similar between the two evaluations. This may be attributed to the fact that parturitions’ family support was lower in T2 compared to that in T1. Indeed, previous studies suggested that social support was a beneficial factor for maternal mental health [41]; one study even suggested that pregnant women who enjoyed their partners’ support were at a decreased risk for depressive symptoms during the pandemic [42]. Nevertheless, as we found no differences in terms of overall social support between T1 and T2, we may deduce that the impact of COVID-19 on maternal mental health is much more complex and does not rely on social and family support alone. It is possible that additional factors that were not evaluated in our study impact maternal wellbeing. Moreover, it is possible that a new component of maternal burnout, attributed to the newly arising maternal challenge of combining working hours with homeschooling and increased housework chores (given the reduced potential for outsourcing domestic services such as cleaning, cooking, etc.), also contributed to increased maternal PPD symptoms over time [43].

To the best of our knowledge, this is the first study to examine the longitudinal effects of COVID-19 on PPD symptoms. As a multicenter cohort study, we were able to reach a diverse population in terms of ethnicity, religion, and socioeconomic status. The exclusion of women who delivered before 34 gestational weeks minimized the bias of the development of psychopathology symptoms due to prematurity [44]. Unlike other studies (16–18), we did not use social media to recruit patients, thus decreasing the bias associated with the accessibility to technology or the involvement with social networks. In addition, at least some of the maternal reports on the demographics, obstetric, and delivery data were validated with their medical records. Lastly, in T1, we specifically approached women about 10 weeks after delivery, which is the acceptable time interval for the detection of postpartum-related psychopathologies. T2 was performed approximately 6 months after delivery, allowing us to examine the parturients’ mental health over time.

Nevertheless, our study is not free of limitations. A substantial number of women did not answer the first recruitment phone call, which reflects a response bias, though it is perhaps less significant compared to approaching participants through social media. The dropout between T1 and T2 may also have influenced the results of our study. However, we found no differences in the severity of PPD symptoms between the two groups in T1. Moreover, women who did not drop out of the study were significantly older and were less economically preoccupied than their counterparts. These characteristics may have allowed them to be at their leisure to participate in the study. Additionally, the mean timing of T1 was at 11 weeks postpartum, which may have also influenced results, and there may be a self-selection bias, as not all women who consented to T2 responded to 70% of the questionnaires. Lastly, we should acknowledge that there may be, as in all of the self-report questionnaires, a subjective nature of the patients’ descriptions of the impact of the birth experience on their feelings, as we base our results on online questionnaires without in-person psychological evaluation of the parturients. Further qualitative research on a larger-scale population for generalizability could broaden our knowledge and present a more complete picture regarding parturients’ birth experiences during the COVID-19 pandemic.

The fact that PPD symptoms were not reduced and even exacerbated over time should raise concerns for caregivers who treat women who gave birth during the COVID-19 era. It is evident that this particular sub-population of parturients during COVID-19 restrictions and strict regulations had unique childbirths and early motherhood experiences [17], we must note that, although time passes and the general population adapts to restrictions and economic and social changes, parturients may still experience increased emotional distress, which may lead to the development of PPD. Further research is needed to understand how to optimally treat and reduce PPD symptoms among these women. Clinicians treating parturients should be aware of the increased risk for PPD during the pandemic and refer women who experience symptoms to receive medical and emotional support.

Unfortunately, the COVID-19 era is not behind us. Although worldwide vaccination has begun, many countries are still under COVID-19 restrictions and regulations, facemasks are still mandatory in many regions, social distancing is still enforced, and many households have not fully recovered from the economic crisis. The general population, especially women, are under stress owing to the pandemic, having to cope with increased responsibilities at home and at work, which may cause even more mental distress [43]. It is quite possible that these women who present PPD symptoms that, in ordinary times, would have decreased over time are now experiencing an increase in symptoms considering the ongoing stress, lockdowns, and general distress, meaning that the COVID-19 pandemic perpetuates the depressive symptoms of parturients. Parturients are prone to develop psychopathologies to begin with [45,46], which in turn may affect their relationship with the newborn [47] and may also affect the emotional and social development of the neonate [48]. Since the long-term consequences of the pandemic on parturients’ mental health are not entirely clear, we must continue to investigate the longitudinal effects of the pandemic on the mental wellbeing of women who gave birth during this period. In addition, as caregivers, we must be aware that women who have given birth during this period are at an increased risk of developing psychopathologies, and we must try to identify the symptoms at an early stage in order to offer support, guidance, and, if necessary, appropriate mental therapy through various platforms [49].

## Figures and Tables

**Figure 1 jcm-11-03488-f001:**
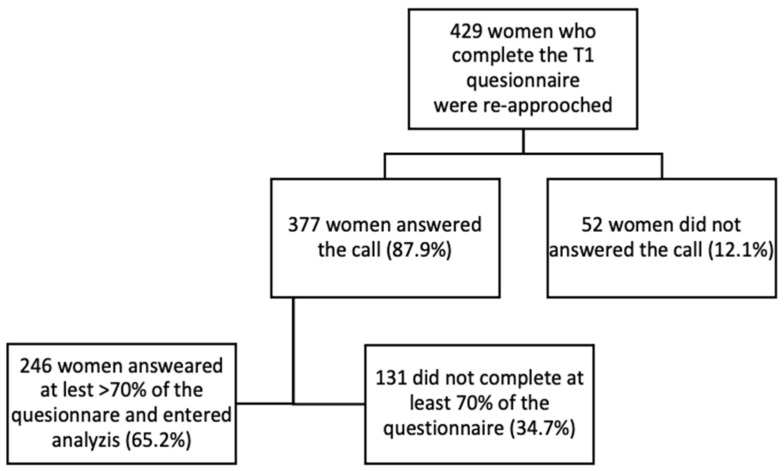
Study cohort.

**Figure 2 jcm-11-03488-f002:**
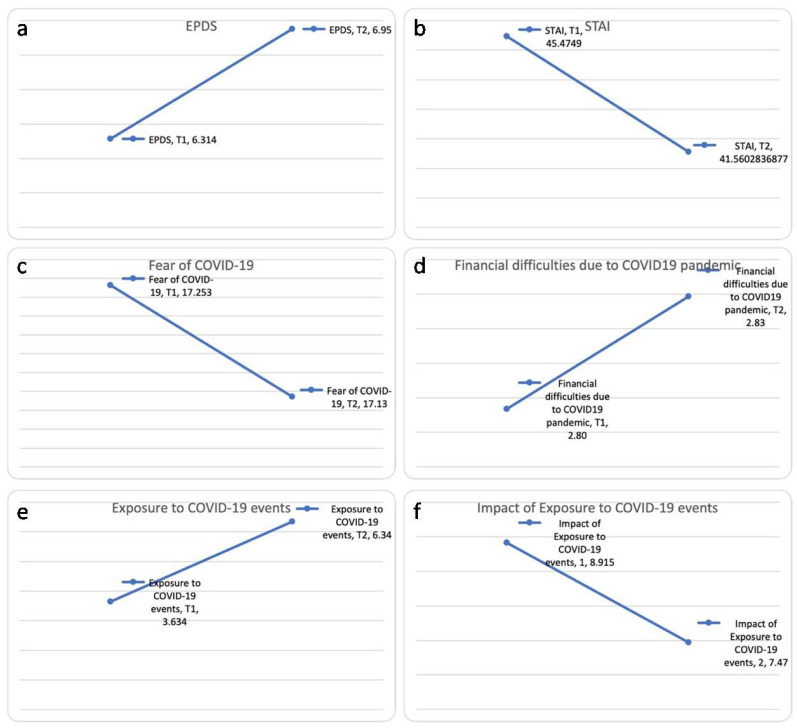
The difference between the EPDS scores (**a**), STAI scores (**b**), fear of COVID-19 (**c**), financial difficulties due to COVID-19 pandemic (**d**), exposure to COVID-19 events (**e**), and impact of exposure to COVID-19 events (**f**) between T1 and T2.

**Table 1 jcm-11-03488-t001:** Maternal characteristics and clinical measurements of the cohort group.

	T1, *n* = 246	T2, *n* = 246
**Maternal characteristic**		
Maternal age, years	32.14 ± 5.1	
Nulliparity	65 (26.4%)	
Ethnicity:		
• Jewish	186 (75.6%)	
• Arabic	60 (24.4)	
Religious level (scale 1–5: 1—secular to 5—very religious)	1.77 ± 0.92	
Marital Status:		
• Married	219 (89%)	
• In a relationship	10 (4.5%)	
• Separated/single	16 (6.5%)	
Education:		
• Elementary	1 (0.4%)	
• High school	78 (31.7%)	
• Bachelor’s degree	105 (42.7%)	
• Master’s degree	55 (22.4%)	
• Doctorate	7 (2.8%)	
**Delivery characteristics**		
Gestational age at delivery	39.50 ± 1.1	
Neonatal birth weight	3286.86 ± 11.0	
Delivery mode		
Vaginal delivery	198 (80.5%)	
Cesarean section	48 (19.5%)	
**Maternal mental health and confounders**		
Delivery to questionnaire interval, weeks	11.03 ± 1.6	28.48 ± 2.1
EPDS ^@^ (scale 0–30)	6.31 ± 5.6	6.92 ± 5.9
Financial difficulties due to the COVID-19 pandemic (scale 1–5: 1—“not worried at all”, 5—“worried all the time”).	2.80 ± 1.2	2.83 ± 1.1
Exposure to COVID-19 events, number of exposures	3.63 ± 1.8	6.34 ± 2.3
Impact of exposure to COVID-19 events	8.91 ± 4.6	7.47 ± 4.1
Fear of COVID-19 (scale 7–35)	17.25 ± 5.3	17.10 ± 5.3
STAI ^#^ (scale 20–80)	45.35 ± 16.4	41.47 ± 14.0
Total score MSPSS ^$^ (scale 1–7)	5.99 ± 1.1	5.88 ± 1.2
• MSPSS ^$^–significant other (scale 1–7)	6.30 ± 1.2	6.24 ± 1.2
• MSPSS ^$^–family (scale 1–7)	6.10 ± 1.3	5.91 ± 1.4
• MSPSS ^$^–friends (scale 1–7)	5.58 ± 1.5	5.48 ± 1.7

For categorical variables, the results are presented as the value (%), and for continuous variables, the results are presented as the value ± standard deviation (SD). ^@^ EPDS—Edinburgh Postnatal Depression Scale; ^#^ STAI—State Anxiety Inventory (STAI) questionnaire; ^$^ MSPSS—Multidimensional Scale of Perceived Social Support.

**Table 2 jcm-11-03488-t002:** Paired samples test: comparing the results of the T1 and T2 questionnaires.

T1 to T2 Delta	Mean	Std. Deviation	Std. Error Mean	95% Confident Interval		t	df	Sig.
				Lower	Upper			
EPDS ^@^	−0.64	4.59	0.29	−1.21	−0.06	−2.17	244	**0.031**
Exposure to COVID-19 events	−2.71	2.50	0.16	−3.02	−2.39	−16.98	245	**0.000**
Impact of exposure to COVID-19 events	1.45	4.91	0.31	0.83	2.06	4.62	245	**0.000**
Fear of COVID-19	0.12	4.43	0.28	−0.44	0.67	0.42	244	0.676
STAI ^#^	3.91	13.67	0.89	2.16	5.67	4.39	234	**0.000**
MSPSS ^$^ total score	0.11	0.85	0.06	−0.01	0.22	1.87	219	0.062
MSPSS ^$^ significant other	0.06	1.08	0.07	−0.09	0.20	0.78	219	0.438
MSPSS ^$^ family	0.17	0.92	0.06	0.04	0.29	2.68	216	**0.008**
MSPSS ^$^ friends	0.10	1.31	0.09	−0.08	0.27	1.10	219	0.271
Financial difficulties due to the COVID-19 pandemic	0.00	0.34	0.02	−0.03	0.05	0.37	245	0.706

Significant *p* values (<0.05) are in bold. ^@^ EPDS—Edinburgh Postnatal Depression Scale; ^#^ STAI—State Anxiety Inventory (STAI) questionnaire; ^$^ MSPSS—Multidimensional Scale of Perceived Social Support.

**Table 3 jcm-11-03488-t003:** Test of within-subjects effect: factors contributing to the changes in the EPDS ^@^ scores between T1 and T2.

	Sum of Squares	df	Mean Square	F	Sig.
Delta of MSPSS ^$^ family (T2-T1)	38.401	1	38.401	4.308	**0.039**
Delta of exposure to COVID-19 events (T2-T1)	6.756	1	6.756	0.758	0.385
Delta of impact of exposure to COVID-19 (T2-T1)	44.908	1	44.908	5.038	**0.026**
Delta of STAI ^#^ (T2-T1)	285.135	1	285.135	31.988	**0.000**

^@^ EPDS—Edinburgh Postnatal Depression Scale; ^#^ STAI—State Anxiety Inventory (STAI) questionnaire; ^$^ MSPSS—Multidimensional Scale of Perceived Social Support.

## Data Availability

The data are available from the corresponding author on request.

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
