# Peer review of "Postpartum Depression in COVID-19 Days: Longitudinal Study of Risk and Protective Factors"

_jcm, 2022, doi:10.3390/jcm11123488_

Round 1

Reviewer 1 Report

This study has some notable strengths: examines a population that needs more research (PPD during pandemic), especially given the long-term effects on pregnant individual, child, and family; a longitudinal design; and there are concise lockdown dates used for context. However, there are some significant missing components from the manuscript and the Discussion overstates the findings. Revisions to the manuscript are needed.

  • Introduction
    • Nice summary, more needs to be said specifics that are known about PPD and COVID-19. Most of the literature was on COVID-19 but what is known about how it affected perinatal women specifically? Why are perinatal women especially vulnerable? I would recommend cutting down information on COVID-19 broadly affecting the world and focus on how it affected perinatal women.
    • Hypothesis should be clarified-do you expect higher scores in Time 1 compared to Time 2? It is unclear from this introduction.
  • Method
    • PPD can be experienced sooner than 10 weeks after delivery – more information on why this time frame was chosen over 4 weeks (DSM-5 threshold) or 6-8 weeks (other common research and clinical thresholds).
    • The exact variables are described in the measures, so Data Collection is redundant. This could be better used as defining outcome variables and covariates of interest and be much shorter.
    • Measures-indicate if higher scores indicate more fear, severity, etc.
    • Include psychometrics in measures (validity, reliability, especially for newer COVID measures)
  • Results
    • Define economically preoccupied
    • The number of women who were contacted, consented, etc., should go in the procedure part of the Methods, not the Results.
  • Discussion
    • Small response rate for those answering questionnaires. Can this be addressed further? It seems smaller than expected. Why may women not have wanted to participate?
    • The average EPDS score is below clinical threshold for depression (10) and there is a very small increase in the number of women who are above 10 on the EPDS (57-66). Was this a statistically significant change? Also, need to revise generalizations because there are fairly comparable numbers at T1 and T2.
    • The authors’ main message is somewhat muddled. EPDS scores continue to be low, anxiety scores decrease, so how is this a heightened time for PPD?

Author Response

Editor in Chief
Journal of Clinical Medicine

Dear Editor,

Thank you for considering our manuscript for publication in the "JCM" journal. We would like to thank the reviewers for their valuable opinions and suggestions. Please find attached our revised manuscript entitled " Longitudinal study of Postpartum Depression in COVID-19 Era: Risk and Protective Factors". We have proceeded with the revisions and have enclosed an itemized list of the changes.

Please do not hesitate to contact us if you have any further questions or concerns.

Sincerely,

Rinat Gabbay-Benziv, MD

Department of Obstetrics and Gynecology

Hillel Yaffe Medical Center

Hadera, Israel

Tel: +972-4-6304313

Email: gabbayrinat@gmail.com

Reviewer 1:

This study has some notable strengths: examines a population that needs more research (PPD during pandemic), especially given the long-term effects on pregnant individual, child, and family; a longitudinal design; and there are concise lockdown dates used for context. However, there are some significant missing components from the manuscript and the Discussion overstates the findings. Revisions to the manuscript are needed.

Introduction

  • Nice summary, more needs to be said specifics that are known about PPD and COVID-19. Most of the literature was on COVID-19 but what is known about how it affected perinatal women specifically? Why are perinatal women especially vulnerable? I would recommend cutting down information on COVID-19 broadly affecting the world and focus on how it affected perinatal women.

Answer: We added more information focusing on the specific impact of COVID-19 on postpartum women:

“After birth, both familial and hormonal changes have a substantial impact on women’s mental health, which may aggravate existing psychopathologies or cause the development of specific psychopathology such as postpartum depression (PPD).

…. Pregnant women experienced higher levels of fear of COVID-19 infection that was further augmented by fear of endangering the fetus. Additionally, social distancing and lockdowns limited maternal social support networks, as well as access to health care services; This in turn has increased maternal risk for the development of psychological disorders. Moreover, the use of personal protective equipment was found to be an independent factor for developing depressive and post-traumatic stress symptoms during the postpartum period.”

  • Hypothesis should be clarified-do you expect higher scores in Time 1 compared to Time 2? It is unclear from this introduction.

Answer: We thank the reviewer for this comment. Since COVID-19 pandemic was an unprecedentedly significant novel event and had a widespread international impact, it was not possible to predict the response of women over time. On the one hand, all mentioned sequela of pandemic has worsened maternal depression, however, on the other hand, over time, women may have become accustomed to the new situation.  Writing the manuscript, it was hard for us to predict the overall effect therefore we did not state a specific hypothesis. Our aim, in this study was to examine the longitudinal trend of PPD symptoms presentation.

Method

  • PPD can be experienced sooner than 10 weeks after delivery – more information on why this time frame was chosen over 4 weeks (DSM-5 threshold) or 6-8 weeks (other common research and clinical thresholds).

Answer: We thank the reviewer for this comment. We chose this time frame as this is the time frame accepted in Israel (4-10) for PPD assessment. In fact, although the mean T1 interval from delivery was 11.03±1.6 the range started from 7.03 weeks.  In the methods section, we explained the choice to approach women 10 weeks after delivery: “this is the acceptable time interval from birth that is used to assess PPD or other birth-related psychopathologies” (Reference in the manuscript). We acknowledge the reviewer comment and will add this in the limitation section.

  • The exact variables are described in the measures, so Data Collection is redundant. This could be better used as defining outcome variables and covariates of interest and be much shorter.

Answer: We thank the reviewer for this comment and shortened the Data collection paragraph.  

  • Measures-indicate if higher scores indicate more fear, severity, etc.

Answer: Done.

  • Include psychometrics in measures (validity, reliability, especially for newer COVID measures)

Answer: Done.

Results

  • Define economically preoccupied

Answer: Economically preoccupied refers to the scale “Financial difficulties due to COVID19 pandemic”. We added an additional explanation in the Methods section.

  • The number of women who were contacted, consented, etc., should go in the procedure part of the Methods, not the Results.

Answer: We thank the reviewer for this comment, however we feel that in the methods our approach and plan should be defined and the result section should include the final description of women recruited. Nevertheless, if the editor will think that these numbers should appear in the methods section, we will change it accordingly.

Discussion

  • Small response rate for those answering questionnaires. Can this be addressed further? It seems smaller than expected. Why may women not have wanted to participate?

Answer: We thank the reviewer for this comment. We included in the text an explanation referring to the dropout of participants from the study: “The dropout between T1 and T2 may have influenced the results of our study. However, we found no differences in the severity of PPD symptoms between the two groups in T1. Moreover, women who did not drop out of the study were significantly older and were less economically preoccupied than their counterparts. These characteristics may have allowed them to be at their leisure to participate in the study.”

  • The average EPDS score is below the clinical threshold for depression (10) and there is a very small increase in the number of women who are above 10 on the EPDS (57-66). Was this a statistically significant change? Also, need to revise generalizations because there are fairly comparable numbers at T1 and T2.

Answer: Indeed, the cut-off for abnormal EPDS differs between studies. While some uses cutoff of 10 (referenced in the manuscript), other uses higher scores as cutoff. As a routine in Israel, postpartum women are asked to answer the EPDS questionnaire, with a score of 10 or higher raising the suspicion of symptoms of postpartum depression, and further investigation is advised. EPDS can be evaluated both as a categorical variable (with cutoff) or as a continuous variable. In our study, we focused mainly on EPDS as a continuous variable as it is clear that the higher the score is - the higher the chances are for postnatal depression.

Statistically, we conducted a  paired-sample  t-test between the EPDS score in T1 and T2 and found a p-value of 0.031  which means statistical significance.   

The absolute number of women may seem small, but in the percentage aspect- these are differences that cannot be ignored. While 57 of the women in T1 are 23.2 % of the participants, 66 women in T2 are 26.8% of the total women who answered the questionnaire. As for the generalization of the results, the main limitation is the sample size and all dependent variables that confound postpartum depression and may defer in different settings. We added this to the limitations.

  • The authors’ main message is somewhat muddled. EPDS scores continue to be low, anxiety scores decrease, so how is this a heightened time for PPD?

Answer: Depression is not a categorical variable and has meaning in terms of continuity over time. Unlike in routine times, where PPD symptoms are supposed to decrease in frequency and severity as one moves away from birth, our study revealed that in the COVID-19 era PPD symptoms increased over time in terms of severity and frequency. These are issues who need to be addressed.

Reviewer 2 Report

Several major revisions are necessary.

Specific comments:

  1. It has often been thought that pregnancy is protective against the development of depression, primarily because of the lower suicide rate during pregnancy and during the 2 years after giving birth (citation: pubmed.ncbi.nlm.nih.gov/14519602). In contrast, the postpartum time period clearly was a period of increased risk for the development of MDD (citation: pubmed.ncbi.nlm.nih.gov/22860768).
  2. A recent study found pregnancy to be associated with a reduced risk for depressive symptoms during the pandemic (citation: ncbi.nlm.nih.gov/pmc/articles/PMC8072624). This was attributed to increased partner support, healthy behaviors, and positive appraisal of the pregnancy. 
  3. "... longitudinal, multicenter, questionnaire-based, case-control, observational study" - how is this a case-control study?
  4. More details on the study exclusion and inclusion criteria are necessary. Were mothers with multiple pregnancies e.g. twins or triplets eligible to take part in the study? What about mothers of advanced maternal age? What about mothers who had COVID-19 infection?
  5. A CONSORT type diagram would be helpful. Please demonstrate the total number of pregnant women who delivered during the two time periods and if there were any women excluded from the analyses.
  6. Please update the description for Table 2. "Paired samples test: T1 and T2" is neither informative nor descriptive.
  7. Did the authors adjust for baseline depression or anxiety as a covariate? Additionally, socioeconomic status still varies over time in this age range.
  8. The authors should make more concrete suggestions in the discussion section. As resources could be particularly scarce during a serious pandemic situation, timely psychological support could also take many forms, including telemedicine and informal support groups (citation: pubmed.ncbi.nlm.nih.gov/32380875). This should be mentioned.
  9. Please change "Data is available by request" to "Data is available from the corresponding author on request".

Author Response

Editor in Chief
Journal of Clinical Medicine

Dear Editor,

Thank you for considering our manuscript for publication in the "JCM" journal. We would like to thank the reviewers for their valuable opinions and suggestions. Please find attached our revised manuscript entitled " Longitudinal study of Postpartum Depression in COVID-19 Era: Risk and Protective Factors". We have proceeded with the revisions and have enclosed an itemized list of the changes.

Please do not hesitate to contact us if you have any further questions or concerns.

Sincerely,

Rinat Gabbay-Benziv, MD

Department of Obstetrics and Gynecology

Hillel Yaffe Medical Center

Hadera, Israel

Tel: +972-4-6304313

Email: gabbayrinat@gmail.com

Reviewer 2:

  1. It has often been thought that pregnancy is protective against the development of depression, primarily because of the lower suicide rate during pregnancy and during the 2 years after giving birth (citation: pubmed.ncbi.nlm.nih.gov/14519602). In contrast, the postpartum time period clearly was a period of increased risk for the development of MDD (citation: pubmed.ncbi.nlm.nih.gov/22860768).
  2. A recent study found pregnancy to be associated with a reduced risk for depressive symptoms during the pandemic (citation: ncbi.nlm.nih.gov/pmc/articles/PMC8072624). This was attributed to increased partner support, healthy behaviors, and positive appraisal of the pregnancy. 

Answer: We thank and appreciate this response. As we wrote in our response to the first reviewer - it was hard to predict the longitudinal effect of the pandemic as so many variables were disrupted at different directions in an unprecedent way. We added some of these references to the.  

  1. "... longitudinal, multicenter, questionnaire-based, case-control, observational study" - how is this a case-control study?

Answer: We thank the reviewer for this important note. We changed the description of the study: this was a longitudinal, multicenter, cohort analysis.

  1. More details on the study exclusion and inclusion criteria are necessary. Were mothers with multiple pregnancies e.g. twins or triplets eligible to take part in the study? What about mothers of advanced maternal age? What about mothers who had COVID-19 infection?

Answer: Parturients younger than 18 years or who delivered earlier than 34 gestational weeks were excluded from the study. Initially we addressed women who delivered during the time frame mentioned in the text, also to mothers with multiple gestation, mothers with advanced maternal age and mothers who had COVID-19 infection.

The final analysis included only women who answered 70% of the questionnaire and therefore did not include women with multiple pregnancies who were initially scant. In terms of women's age there was no age limit, the oldest patient in the analysis was 46 years old. We also included women infected with COVID-19. In total, only 5 women completed the T2 questionnaire and had COVID-19 infection (It should be considered that these were the first months of the pandemic in Israel). They reported this as part of the “COVID-19 exposure questionnaire “.  

  1. A CONSORT type diagram would be helpful. Please demonstrate the total number of pregnant women who delivered during the two time periods and if there were any women excluded from the analyses.

Answer: Done.

  1. Please update the description for Table 2. "Paired samples test: T1 and T2" is neither informative nor descriptive.

Answer: Done. “Table 2: Paired samples test: Comparing results of T1 and T2 questionnaires.”

  1. Did the authors adjust for baseline depression or anxiety as a covariate? Additionally, socioeconomic status still varies over time in this age range.

Answer: When examine the T2 population, we took into consideration any maternal psychiatric background, deduced by report of psychiatric drugs and by the parameter. Since only 5 participants used psychiatric drugs, this parameter didn’t enter the analysis as a covariant.

Regarding socioeconomic status - it is unlikely that in three months the one’s economic status will change dramatically. Nevertheless, since the pandemic had a huge impact on the economic condition of many, we created the "Financial difficulties due to COVID19 pandemic" variable, to which we refer both in the text and in the final analysis. No differences were found in this variable between T1 and T2 (Table 2).  

  1. The authors should make more concrete suggestions in the discussion section. As resources could be particularly scarce during a serious pandemic situation, timely psychological support could also take many forms, including telemedicine and informal support groups (citation: pubmed.ncbi.nlm.nih.gov/32380875). This should be mentioned.

Answer: Done. ”In addition, as caregivers, we must be aware that women who have given birth during this period are at increased risk of developing psychopathologies and we must try to identify the symptoms at an early stage to offer support, guidance and if necessary appropriate mental therapy, through various platforms.” Reference was added.

  1. Please change "Data is available by request" to "Data is available from the corresponding author on request".

Answer: Done

Round 2

Reviewer 2 Report

Thanks for the revisions.

Still needs a close edit for language and grammar.